# The Combination of Buparvaquone and ELQ316 Exhibit a Stronger Effect than ELQ316 and Imidocarb Against *Babesia bovis* In Vitro

**DOI:** 10.3390/pharmaceutics16111402

**Published:** 2024-10-31

**Authors:** Natalia M. Cardillo, Nicolas F. Villarino, Paul A. Lacy, Michael K. Riscoe, Joseph Stone Doggett, Massaro W. Ueti, Chungwon J. Chung, Carlos E. Suarez

**Affiliations:** 1Animal Disease Research Unit, United States Department of Agriculture-Animal Research Unit (USDA-ARS), 3003 ADBF, WSU, Pullman, WA 99163, USA; lacyp@wsu.edu (P.A.L.); massaro.ueti@usda.gov (M.W.U.); chungwon.chung@usda.gov (C.J.C.); carlos.suarez@wsu.edu (C.E.S.); 2Estación Experimental INTA Paraná, Consejo Nacional de Investigaciones Científicas y Técnicas (CONICET), Buenos Aires 2290, Argentina; 3Department of Veterinary Microbiology and Pathology, Washington State University, Pullman, WA 99164, USA; 4Program in Individualized Medicine, Department of Veterinary Clinical Sciences, College of Veterinary Medicine, Washington State University, Pullman, WA 99164, USA; nicolas.villarino@wsu.edu; 5VA Portland Healthcare System, 3710 SW US Veterans Hospital Road, Portland, OR 97239, USAdoggettj@ohsu.edu (J.S.D.); 6Department of Microbiology and Molecular Immunology, Oregon Health & Science University, 3181 SW Sam Jackson Park Road, Portland, OR 97239, USA; 7School of Medicine, Division of Infectious Diseases, Oregon Health & Science University, 3181 SW Sam Jackson Park Road, Portland, OR 97239, USA

**Keywords:** *apicomplexa*, treatment, cytocrome, new drugs, dose-response

## Abstract

**Background/Objectives:** Bovine babesiosis is a vector-borne disease transmitted by ticks that causes important losses in livestock worldwide. Recent research performed on the drugs currently used to control bovine babesiosis reported several issues including drug resistance, toxicity impact, and residues in edible tissue, suggesting the need for developing novel effective therapies. The endochin-like quinolones ELQ-316 and buparvaquone (BPQ) act as cytochrome *bc*1 inhibitors and have been proven to be safe and efficacious against related apicomplexans, such as *Plasmodium* spp. and *Babesia microti*, without showing toxicity in mammals. The objectives of this study are investigating whether ELQ-316, BPQ, and their combination treatment could be effective against *Babesia bovis* in an in vitro culture model and comparing with imidocarb (ID), the routinely used drug. **Methods:** In vitro cultured parasites starting at 2% percentage of parasitemia (PPE) were treated with BPQ, ELQ-316, ID, and the combinations of BPQ + ELQ-316 and ID + ELQ-316 at drug concentrations that ranged from 25 to 1200 nM, during four consecutive days. The IC50% and IC99% were reported. Parasitemia levels were evaluated daily using microscopic examination. Data were compared using the non-parametrical Mann–Whitney and Kruskall–Wallis test. **Results:** All drugs tested, whether used alone or in combination, significantly decreased the survival (*p* < 0.05) of *B. bovis* in in vitro cultures. The combination of BPQ + ELQ-316 had the lowest calculated inhibitory concentration 50% (IC50%) values, 31.21 nM (IC95%: 15.06–68.48); followed by BPQ, 77.06 nM (IC95%: 70.16–86.01); ID + ELQ316, 197 nM (IC95%:129.0–311.2); ID, 635.1 nM (IC95%: 280.9–2119); and ELQ316, 654.9 nM (IC95%: 362.3–1411). **Conclusions:** The results reinforce the higher efficacy of BPQ at affecting *B. bovis* survival and the potential synergistic effects of its combination with ELQ-316, providing a promising treatment option against *B. bovis*.

## 1. Introduction

Babesiosis, primarily caused by *B. bovis* and *B. bigemina*, is a tick-borne parasitic disease that significantly impacts the cattle industry worldwide [1,2,3,4]. The disease can lead to acute disease and persistent infection in livestock [5], causing high morbidity and mortality in tropical and semi-tropical regions [6].

Effective control of piroplasmosis involves three primary strategies: vaccination, the use of antipiroplasm drugs, and vector control measures [7,8,9,10,11,12]. Chemotherapy is a critical tool for babesiosis control, with several drugs and combinations proving to be effective [13,14,15,16,17,18,19]. However, most, if not all, of the more effective drugs currently in use have toxic side effects or accumulation of residues in the edible tissues of treated animals or generate drug-resistant parasites [20,21,22,23]. These outcomes triggered the urgent need to develop novel antipiroplasm drugs and effective new regimens [9,24,25].

Imidocarb is currently the first-line treatment for bovine babesiosis, but it is not yet approved for use in cattle in the US [26]. This drug has potent babesiacidal effects, providing clearance of parasites and prophylactic protection [7,13]. However, ID’s mechanism of action is not well understood. ID has toxic effects in animals and persistent residues in edible tissues [7,8,27,28,29].

BPQ, a second-generation hydroxynaphthoquinone antiprotozoal drug, selectively inhibits the parasite’s Qo quinone-binding site of the mitochondrial cytochrome *b* electron transport system, leading to its lethal effect on parasites [30,31,32,33]. Although BPQ is registered in around 20 countries for treating East Coast fever and tropical theileriosis [34], its approval in other countries is limited due to residual toxicity concerns [35].

BPQ was initially developed as an anti-malarial drug and has also shown promise in treating *Theileria* species in previous studies [34,36,37,38,39,40,41,42]. Furthermore, it was recently reported that BPQ is significantly more effective than ID in inhibiting the growth of *B. bovis* in vitro, at various concentrations and time points [19].

ELQ-316 is an endochin-like quinolone (ELQ) compound that selectively inhibits the parasite’s Qi quinone-binding site of mitochondrial cytochrome *bc*1 and was highly effective against *Plasmodium falciparum*, *Babesia microti*, and *Toxoplasma gondii* [43]. It was also shown to be effective in in vitro treatment of *Theileria* spp. and *Babesia* spp. parasites [16]. The ELQs also demonstrated high antimalarial potency in vitro and in vivo, parasite selectivity, chemical and metabolic stability, desirable pharmacokinetics, and low mammalian cell toxicity. In addition to their antimalarial activity, compounds in the ELQ series were later found to be highly active against other Apicomplexa, for which effective treatments are urgently needed. With favorable properties and broad-spectrum activity, the ELQ compound class may yield effective, safe treatments for a range of important human and animal afflictions.

Although a single-drug treatment is preferred, the use of mitochondrial *bc*1 inhibitors may result in the emergence of resistant parasites associated with a mutation in the Cytb Qi or Qo sites. However, the combination of compounds with activity against both sites (Cytb Qi and Qo), like ELQ-316 and BPQ, may improve the treatment and reduce the possibility for the emergence of parasites with drug resistance [44]. This was previously demonstrated by complete and persistent clearance of parasites [45]. Therefore, BPQ and ELQ-316 may offer effective options for managing babesiosis, especially in cases where traditional treatments like imidocarb dipropionate fail, result in frequent clinical relapses, or induce toxicity or generate undesired residues in food. As far as we know, there are no previous studies regarding the effects on the survival of these combinations against *B. bovis.*

The aim of this study was to compare the effects on *B. bovis* in vitro survival of ELQ-316, BPQ, ID, and the combinations of ID + ELQ-316 and BPQ + ELQ-316.

## 2. Materials and Methods

### 2.1. Compounds Tested

BPQ (98% purity) was obtained from Combi-Blocks, Inc. (San Diego, CA, USA). ID (VETRANAL^TM^, Supelco^®^ Buchs, Switzerland) was used as a positive control for the in vitro inhibition assays for *B. bovis*, using an identical protocol to BPQ described below. The purity of ID was determined to be >98% by proton-nuclear magnetic resonance spectroscopy and high-performance liquid chromatography (HPLC), according to the certificate of analysis.

ELQ-316 was synthesized by following methods previously described by Nilsen et al. (2014), identified by proton nuclear magnetic resonance (1H NMR), and determined to be ≥95% pure by reversed-phase high-performance liquid chromatography (RP-HPLC) [46].

BPQ, ELQ-316, and ID were diluted in 100% dimethyl sulfoxide (DMSO) to prepare stock solutions, which were kept at room temperature until use. Working solutions were freshly prepared in a parasite culture medium every test day before being added to the parasite cultures.

### 2.2. Babesia bovis In Vitro Culture

*B. bovis* (Texas T2Bo strain) [47] were grown in long-term microaerophilic stationary-phase cultures and incubated at 37 °C in an atmospheric condition of 5% CO_2_, 5% O_2_, and 90% N_2_, as previously described [48]. *B. bovis* were cultured in 96-well culture plates, containing 180 µL per well of PL culture media [pH 7.2, to prepare 100 mL, 29 mL F-12K Nutrient Mixture + L-glutamine (Life Technologies, Carlsbad, CA, USA), 29 mL Stable Cell IMDM (Sigma Aldrich, St. Louis MO, USA), 2 mL 0.5 M TAPSO (Sigma Aldrich), 0.5 mL Antibiotic Antimycotic solution 100× (Sigma Aldrich, St. Louis, MO, USA), 1 mM calcium chloride (Sigma Aldrich), 100 µL Antioxidant Supplement 1000× (Sigma Aldrich), 1 mL Insulin-Transferrin-Selenium-Ethanolamine 100× (Sigma Aldrich, St. Louis, MO, USA), 1 mL 50% Glucose (Teknova, Hollister, CA, USA), and 500 µL L-glutamine 200 mM (GIBCO, Grand Island, NY, USA)], supplemented with 40% bovine serum and containing a suspension of 10% cells volume of erythrocytes.

### 2.3. In Vitro Growth Inhibitory Assays

The in vitro inhibitory efficacies of BPQ, ELQ-316, and ID against the survival of the T2Bo *B. bovis* strain were evaluated with a starting percentage of parasitized erythrocytes (PPE) of 2%. *B. bovis* was grown, as described above, in culture media containing different concentrations of BPQ, ELQ-316, ID, and the combinations (BPQ + ELQ-316 and ID + ELQ-316) at 25, 50, 75, 100, 150, 200, 300, 600, and 1200 nM concentrations, diluted in 100% DMSO. Parasites cultured in the presence of DMSO (1 µL) and the absence of drug compounds were used as a positive control for parasite growth. Extra wells containing uninfected bovine RBC were prepared and used as negative controls. Parasite cultures were fed daily with fresh culture medium (180 µL/well) containing the respective drug concentration. The experiments were carried out in triplicate wells for each tested concentration and control, over 96 h (4 days). PPE was monitored daily by counting parasites in Hema 3 Stat Pack (Fisher Scientific) stained thin blood smears (GBS). Before the daily change of the media, the supernatant media (180 µL) of each well was collected, and the bottom with the RBC layer was gently mixed; 1 µL of sample was taken from each well to make a thin smear, and the number of infected red blood cells was counted by microscopical examination of 5000 erythrocytes in each slide. The morphological appearance was also observed. The drug responsiveness of the parasites was measured as percent parasitemia after every 24 h exposure to each concentration of the drug until 96 h.

### 2.4. Post-Treatment Survival

At 96 h after the first treatment, fresh medium without drug was replaced in all the culture wells, and 10 μL of fresh RBCs were added. The same procedure was performed for the next five days to determine the post-treatment survival. Quantitative and qualitative parasitemia was determined by microscopic examination of Hema 3 Stat Pack (Fisher Scientific) stained thin blood smears.

### 2.5. Statistical Analysis

Levels of parasitemia were counted every day; the percentage of survival was calculated; and media comparisons between BPQ and BPQ + ELQ-316 and ID and ID + ELQ-316 combinations were conducted through a Mann–Whitney test. Comparison of the mean percentage of survival of *B. bovis* against the three different drug treatments (BPQ, ELQ-316, and ID) was performed using a Kruskall–Wallis test.

The doses of a drug that produced 50% inhibition (IC50%) relative to the control population and the maximal inhibitory concentration (IC99%) were estimated for BPQ, ELQ-316, ID, and the combinations, at concentrations ranging from 25 to 1200 nM, in 24 h, 48 h, 72 h, and 96 h post-incubation. Total inhibitory concentrations (IC99) were determined as the drug doses needed to reduce parasite growth to the same level observed in non-infected erythrocytes (approximately 0.1%). The survival curves were compared using a log-rank (Mantel–Cox) test. The level of significance was set at <0.05. GraphPad Prism ver. 7 software for Windows (Graphpad Software Inc., San Diego, CA, USA) was used for the statistical analysis.

## 3. Results

### 3.1. Comparative In Vitro Survival Effects of BPQ, ID, ELQ-316, BPQ+ ELQ-316, and ID+ ELQ-316 Combinations on B. bovis

Parasite survival levels in 96 h of in vitro cultures in the presence of BPQ, ELQ-316, ID, and the combination BPQ + ELQ-316 and ID + ELQ-316, after starting with 2% PPE, are depicted in Figure 1. The differences in the mean percentages of survival between BPQ, ID, and ELQ-316 are presented in Table 1. Between 75 to 300 nM, BPQ had a significantly higher effect than ELQ-316 in eliminating the parasite under the tested experimental conditions. In contrast, ID and ELQ-316 had similar parasiticidal kinetics but parasites seemed to have a slower response, at the same drug concentrations. ELQ-316 had a significantly higher effect on eliminating parasites in vitro than ID and BPQ at 600 nM, but no differences were seen at 1200 nM between drugs.

Comparisons between BPQ alone and the combination of BPQ + ELQ-316 are shown in Table A1 (Appendix A). While BPQ alone caused a gradual decrease in survival, its combination with ELQ-316 dramatically decreased the parasite survival at 50 nM (Figure 1). The data presented in Figure A1 (Appendix A) also depict the different antiparasitic behavior of BPQ, BPQ + ELQ-316, and ELQ-316, demonstrating a noticeable antiparasitic improvement of the drug combination over the single drug treatments. The calculated survival percentage values were very small in terms of parasitemia (less than 0.05%), and the data could be experimentally biased because the differences between them were minimal. Consequently, the significant difference found at 600 nM was not emphasized in the test because there was no relevant difference in the parasite counts between the two drugs beyond 150 nM.

Comparative parasite survival studies demonstrated that the combination ELQ316+ID also had improved the effects of both drugs separately Table A2 (Appendix A), but the improvement was more gradual and became statistically significant at higher concentrations of both drugs (75 nM), in comparison with the combination with BPQ. At 75 nM and higher concentrations, this combination was significatively more effective than ID alone, showing less than 1% of parasite survival at 1200 nM treatment. Similar results were observed between treatments with ELQ-316 alone and the combination ELQ316+ID, resulting in the most significant decrease in parasite survival at 600 nM (Table A2—Appendix A). The data in Figure A2 (Appendix A) present the different survival percentages of ID, ELQ316+ID, and ELQ316 throughout time, showing the clear improvement through combination treatment in comparison with ID alone.

### 3.2. Drug Potency

The IC50% and IC99% were calculated to compare the drug potencies on the growth of in vitro *B. bovis* T2Bo strain cultures. The combination of BPQ + ELQ-316 had the lowest IC50% of all of the drugs and combinations tested (Table 2). BPQ had the second lowest IC50% but was almost two-fold higher when compared with the combination of BPQ + ELQ-316. No differences were observed between the IC50% of ID and ELQ-316, but those values were around three times the IC50% of the ID + ELQ-316 combination.

Consistent with the IC50 results, the combination BPQ + ELQ-316 had the lowest IC99% (Table 2).

### 3.3. Time and Concentration of Drugs to Reach 0% Survival after Treatment

The ability of parasites to survive in in vitro cultures after drug treatments of different lengths and doses for each drug and combination are presented in Table 3. Different treatments with BPQ, BPQ + ELQ316, ELQ316, and ID + ELQ316 treated parasites were no longer viable, reaching 0% survival at different doses and times post-incubation without drug(s). BPQ seemed to be superior to ELQ316 and ID in terms of time and concentration required to reach 0% survival. The parasites were no longer viable after 1 and 2 days of culture without BPQ in cultures treated previously with 150 nM and 200 nM, respectively. The BPQ + ELQ316 combination had a superior effect to ELQ316 + ID on reaching the time to attain 0% survival. BPQ + ELQ316 combination had a superior effect to ID + ELQ316 on reaching the time to attain 0% survival. The parasites were no longer viable after 1, 2, and 3 days in BPQ + ELQ316-free media, in cultures that were treated previously with drug concentrations above 200 nM, 100 nM, and 50 nM, respectively. Regarding the combination of ID + ELQ316, no parasites were found after 1 day in ID + ELQ316-free media, in cultures treated previously with concentration 600 nM, and after 4 days with concentrations of 300 nM. In cultures treated previously with ID, 0% survival was reached after 4 days in ID-free media, with drug concentrations of 300 nM. No parasites were found after 1 day in ELQ316-free media cultures treated previously with drug concentrations of 600 nM. At 1200 nM, there were no significant differences in timing to reach 0% survival among different drugs in drug-free treatment cultures.

BPQ seemed to be superior to ELQ-316 and ID in terms of the time and concentration required to reach 0% survival. The parasites were no longer viable after 1 and 2 days in BPQ-free media in cultures that were treated previously with drug concentrations above 200 nM and 150 nM, respectively. The combination BPQ + ELQ-316 seemed to have a superior effect to ELQ-316 + ID on reaching the time to attain 0% survival. After the BPQ + ELQ316 combination treatment, parasites were no longer viable after 1, 2, and 3 days of incubation in drug-free culture media from lower concentrations (from 200 nM, 100 nM, and 50 nM, respectively). Regarding ID, 0% survival could be demonstrated only after 4 days of culture using drug-free media when cultures were previously treated at a concentration of 300 nM or higher. No parasites were found after 1 day without ELQ316 treatment at the concentration of 600 nM. With the combination ID + ELQ316, no parasites were found after 1 day without treatment at the concentration of 600 nM and after 4 days at the concentration of 300 nM. At 1200 nM, there was no difference in the time to reach 0% survival among drugs.

## 4. Discussion

The shortage of approved and available drugs in most countries for the effective control of bovine babesiosis underscores the need to discover new, effective, and safer chemotherapeutic alternatives. Moreover, it is particularly important to develop alternative treatments with different mechanisms of action that can be combined to achieve radical cure and to slow down or possibly eliminate the emergence of resistance. The success of treatments depends mainly on early diagnosis, disease severity, dosage and timing of drug treatment, virulence, and phenotypic/genotypic characteristics of the parasite strains involved [49], some of the factors that we tested in this study in vitro with different drugs and combinations.

Herein, we compared the effects of ID, BPQ, and ELQ-316, and the combinations of ID+ ELQ316 and BPQ + ELQ316, on the *B. bovis* post-treatment survival using in vitro cultures. The findings in this study demonstrated that the combination of BPQ + ELQ 316 acted faster than each of the drugs alone, suggesting that the effects of both drugs could be enhanced by combination treatment. This finding is similar to the report on treating *B. microti*-infected immunodeficient mice by Lawres, et al., 2016; increased efficacy of the combination of atovaquone and ELQ-316 was demonstrated with a limited capacity to generate mutations at both the Qo and the Qi sites when exposed to simultaneous drug pressure [45]. Although data in the current study showed that ELQ316 alone did not have better effects killing the parasites than BPQ at lower concentrations, a significant improvement was obtained upon combining both drugs. In addition, we found that ELQ316 could improve BPQ effects, decreasing the parasites’ post-treatment survival from 50 nM, and was similar to BPQ alone after that concentration. As reported previously by Lawres, et al., 2016, through the radical cure of experimental *B. microti* in immunodeficient mice, the superior efficacy of the combination of atovaquone (ATV) and ELQ-316 in eliminating *Babesia* infection is a result of synergism between those drugs or a prolonged half-life of one or both compounds. Moreover, BPQ and ELQ316 combination therapy, acting on different targets of the ubiquinone–cytochrome bc1 complex, is predicted to completely block the electron transport chain through complex III, through a synergism against mitochondrial function, which is essential for parasite survival. Importantly, it was previously demonstrated that ELQ316 was superior to ATV when administered orally to eliminate *T. gondii* infections inside and outside the brain [43,50]. McConnell, et al., 2018, in comparing treatments for toxoplasmosis, suggested that the lower IC50% values of ELQs against *T. gondii* compared with *P. falciparum* may reflect differences in biologic sensitivity to cytochrome bc1 inhibition, structural differences between *T. gondii* and *P. falciparum* cyt bc1, additional targets in *T. gondii*, or differences in the characteristics of the assays despite a high degree of homology between the *T. gondii* and *P. falciparum* cyt bc1 [43]. If *T. gondii* does have greater sensitivity to cyt bc1 inhibition, it may be due *T. gondii’s* use of cyt bc1 for oxidative phosphorylation, while *P. falciparum* primarily relies on cyt bc1 for pyrimidine biosynthesis during its erythrocytic cycle [51]. The same can be applied on *Babesia* parasites; while it is hypothesized that *T. gondii* could readily cease replication and convert to a bradyzoite form because of cyt bc1 inhibition [52], the erythrocytic stages are not known to convert to a metabolically similar quiescent form. It is suggested that parasite-specific structural features of cyt b contribute to susceptibility to the different ELQs [43].

BPQ potency was found with 77.06 nM (IC95%: 70.16–86.01 nM) at 96 h post-treatment, when treated against 2% starting PPE. This value was lower than those reported by Nugraha et al., 2019 (BPQ IC50%: 135 +/− 41 nM) after 96 h of *B. bovis* culture at 1% starting PPE [53] but was comparable to those reported previously by Cardillo, et al., 2024, in *B. bovis* using a lower starting PPE of 1% (IC50%: 50 nM). In agreement with Cardillo et al., 2024, these results show a starting PPE-dependent effect, at least in an in vitro culture model [19]. It remains to be determined, however, whether this observation can be applied to natural infections with fluctuating parasitemia.

ELQ316 and ID showed similar inhibitory effects in all concentrations until 300 nM at 96 h post-infection, and they consistently had comparable potencies (ID: 654.9 nM and 635.1 nM and, respectively). The mechanism of action of ID is not clearly understood [7], but the possible modes of action that have been proposed are related to different targets than ELQ-316 [31]. The ELQ316 potency value against *B. bovis* parasites found in our study was much higher than those reported against *B. duncani* (IC50% = 136.6 1 nM) [54], *B. bovis* (IC50% = 0.07 nM) [16], *Besnoitia besnoiti* (IC50% = 7.97 nM) [55], and *Toxoplasma gondii* (IC50% = 0.66 and 0.35 nM) [56] and (IC50% = 0.007 nM) [50] against the same parasite. Also, according to our study, ELQ316 alone from a 600 nM concentration greatly decreased the parasite survival. Furthermore, ELQ316 also significatively improved the ID effect on the parasites’ post-treatment survival, when used in combination with 100 nM, resulting in the reduction of almost one third of its potency (ID + ELQ316 IC50%: 197 nM). These findings agree with those of Lawres, et al., 2016 [45], who reported that ELQs may prevent recrudescence in immunodeficient mice experimentally infected with *B. microti* [57,58].

ID has been traditionally used with a moderate efficacy and the occurrence of side effects in treating cattle babesiosis. BPQ recently showed to be more effective and faster in killing the parasite than ID in in vitro culture assays [7,8,27]. However, the in vivo effects of these two drugs and long-acting activity could be related to their persistence in cattle tissues for long duration [59]. One of the limitations of this in vitro study is whether the combination of babesicidal drugs lacking antagonistic effects can be a possible way to slow down the emergence of drug resistance, preferably, if the drugs can inhibit the same target receptors at different sites or if they have different mechanisms of action. On the other hand, if these combination treatments could improve both therapeutic potency and efficacy, this could be either by acting synergistically in the animal model and achieving stronger therapeutic effects and/or decreasing the required dose, thereby reducing side effects and the risk of developing drug resistance in the parasites [60]. Previous work performed in the *B. bovis*-related parasite *B. microti* demonstrated that genetic alterations in the Qi binding site of the cytochrome bc1 complex (Cytb) were associated with resistance to ELQ-316 [61]. In addition, Silva et al., 2020, reported full conservation of the two canonical Qo and Qi binding sites of Cytb of the Cytb genes of *B. bovis*, *B. bigemina*, *B. caballi*, and *T. equi* together with the *B. microti* Cytb [16]. Thus, the combination of ELQ316 with BPQ, with action against Qi and Q0 binding sites of the cytochrome bc1 complex, respectively, can potentially help to slow down the onset of resistance or perhaps completely avoid that risk.

When tested using a starting PPE of 2%, the parasites were no longer viable after 1 and 2 days of culture without BPQ in cultures treated previously with 200 nM and 150 nM. These results were identical to those previously obtained by Cardillo, et al., 2024. Regarding ID, 0% survival was demonstrated after 4 days of culture without the drug following the use of 300 nM. The report by Cardillo et al., 2024, did not follow the cultures without ID more than 2 days, but in their study, 0% survival was demonstrated at ID 300 nM in cultures using 0.2% starting PPE, but 1% PPE levels of parasites were still present in cultures after 2 days without the drug. In cultures treated with 600 nM of ELQ316 and the combination of ID + ELQ316, no parasites were found after 1 day of culture without the drug. The same was observed in cultures treated with ID + ELQ316 at 300 nM but after 4 days without the drug. Like our previous report [19], a residual effect against parasites was found after treatment in all drugs and combinations tested. This suggests a persistent effect, likely related to the extent of the impairment or metabolic change that is dependent on the concentrations of each drug used. This is important to consider for the intermittent dose regimen in animals [62].

We found a survival effect depending on the drug doses and times of exposure. We observed a decrease in survival upon longer exposures and a lack of further survival in parasites treated with increased concentrations of drugs. Except for the starting PPE, dose, and time dependence, the effects of the drugs tested herein were mostly in agreement with previous studies on *B. bovis* using endochin-like quinolones, tulathromycin drugs [14,16], BPQ [19], and artemisinin [63].

Overall, the findings in this study suggest that the combination of BPQ + ELQ316 is a promising candidate for in vivo testing as a new babesiacidal regimen. This is supported by several lines of evidence in our study: (1) The potency of BPQ + ELQ316 was 31.21 (IC95%: 15.06–68.48). (2) BPQ+ELQ316 was 100% babesicidal after 1 day in culture after the last treatment with 200 nM concentration. (3) No parasite survival was found after 2 and 3 days of incubation without replacing the drug during daily media changes at doses of 150 nM and 50 nM. (4) ELQ316 improved BPQ’s performance, killing the parasites faster at lower concentration doses than all of the other drugs and combinations tested.

## 5. Conclusions

In summary, all drugs alone or in combinations can significantly affect (*p* < 0.05) *B. bovis* survival. However, the results presented in the current study reinforce the superiority of BPQ in killing *B. bovis* and the potential synergy of its combination treatment with ELQ-316. Our data demonstrate that ELQ-316 in combination with BPQ is highly effective in eliminating parasitemia without relapse after drug discontinuation in vitro for 5 days. The combination treatment of BPQ + ELQ-316 is much more effective than ID + ELQ-316, suggesting increased synergism when both combined drugs target the same pathway. This combination treatment is a promising treatment option that may eliminate clinical *B. bovis* and needs to be tested in the animal model.

## Figures and Tables

**Figure 1 pharmaceutics-16-01402-f001:**
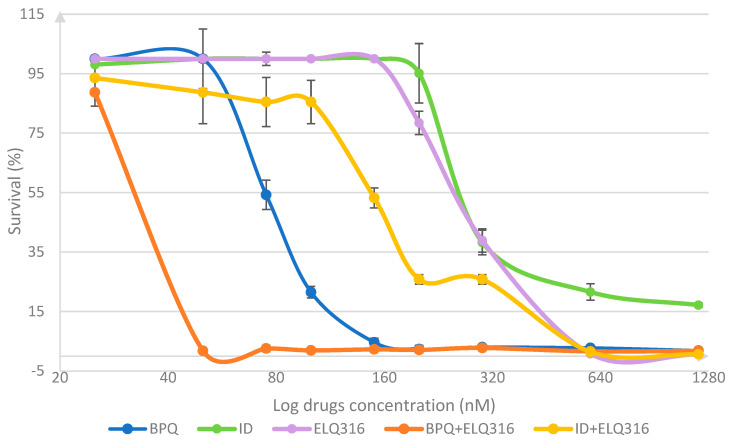
Comparative survival kinetics (mean %) of the in vitro *B. bovis* Texas strain culture during 96 h incubation in the presence of different concentrations of BPQ, ID, ELQ-316, BPQ + ELQ-316, and ID + ELQ-316 combinations, after starting at 2% percentage of parasitemia (PPE). Error bars indicate the standard deviation of the means (n = 3 experiments) for each drug and combination tested.

**Table 1 pharmaceutics-16-01402-t001:** Comparative survival rate (%) of 96 h in vitro *B. bovis* Texas strain culture after starting at 2% PPE in the presence of different concentrations of BPQ, ID and ELQ-316.

Drug Treatment Concentration (nM)	BPQ		ID		ELQ316	
	Mean (%)	CI 95%	Mean (%)	CI 95%	Mean (%)	CI 95%
25	99.87	(99.78–99.96)	98.04	N/A	100	N/A
50	86.6	(77.56–95.64)	99.22	(98.90–99.54)	99.67	(99.45–99.89)
75	54.25 ^a^	(52.33–56.17)	99.22 ^b^	N/A	98.69 ^b^	(97.81–99.57)
100	21.5 ^a^	(20.74–22.26)	100 ^a,b^	N/A	100 ^b^	N/A
150	4.64 ^a^	(4.1–5.18)	100 ^a,b^	N/A	100 ^b^	N/A
200	2.55 ^a^	(2.05–3.05)	89.22 ^b^	(77.98–100)	78.43 ^a,b^	(76.9–79.96)
300	3.01 ^a^	(2.67–3.35)	38.24 ^a,b^	(36.26–40.22)	39.1 ^b^	(38.19–40.01)
600	2.62 ^a^	(2.51–2.73)	21.57 ^a^	(20.25–22.89)	0.65 ^b^	(0.62–0.68)
1200	1.08	(0.88–1.28)	17.16	(16.83–17.49)	0.75	(0.65–0.85)

^a, b^ Drugs with different letters differed significatively (*p* < 0.05). N/A, not applicable: CI were not reported for this concentration’s groups because the mathematical models did not fit the data adequately.

**Table 2 pharmaceutics-16-01402-t002:** Drug potencies (IC50% and IC99%) of different drugs and their combinations on the growth of 96 h in vitro *B. bovis* Texas strain culture after starting at 2% PPE.

Drug	IC50% (nM)		IC99% (nM)	
	Mean	95% CI	Mean	95% CI
BPQ + ELQ-316	31.21	(15.06–68.48)	78	(51.68–117.6)
BPQ	77.06	(70.16–86.01)	186	(97.53–276.9)
ID + ELQ-316	197	(129.0–311.2)	899	(789.2–1009)
ID	635.1	(280.9–2119)	2390	N/A
ELQ-316	654.9	(362.3–1411)	731	(511.1–894.4)

N/A, not applicable: CI were not reported for this concentration’s groups because the mathematical models did not fit the data adequately.

**Table 3 pharmaceutics-16-01402-t003:** Range of concentrations and times in which they reach 0% survival.

Single Drugs and Combination Treatments	Time Post-Treatment Without Drug (h)				
	24	48	72	96	120
Control	N/A				
BPQ	200 to 1200	150 to 1200			
BPQ + ELQ-316	200 to 1200	100 to 1200	50 to 1200		
ELQ-316	600 to 1200				
ID + ELQ-316	600 to 1200		300 to 1200		
ID	1200			300 to 1200	

N/A, not applicable: controls never reached 0% survival during the time of the study (120 h).

## Data Availability

The raw data supporting the conclusion of this article will be made available by the authors, without undue reservation.

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
