# Peer review of "The Combination of Buparvaquone and ELQ316 Exhibit a Stronger Effect than ELQ316 and Imidocarb Against Babesia bovis In Vitro"

_pharmaceutics, 2024, doi:10.3390/pharmaceutics16111402_

Round 1
Reviewer 1 Report
Comments and Suggestions for Authors
see comments in the attachment

Author Response
Comments 1: Abstract: The abstract is organized according to the journal's guidelines. It should be corrected after considering the comments to incorporate the proposed changes. |
Response 1: We appreciate this comment. We reorganized the Abstract according to the Journal guidelines and considered the comments in the resubmitted version as suggested by the Reviewer. The abstract was amended in lines 30 and 32, page 1. Page 1 lines 30 and 32: “The IC50% and IC99% were reported. Parasitemia levels were evaluated daily using microscopic examination. Data was compared using the non-parametrical Mann-Whitney and Krus-kall-Wallis test.” |
Comments 2: Keywords: Suggest replacing the keywords "IC50" by “dose-response”. |
Response 2: Thank you for pointing this out. We agree with this comment. We have replaced “IC 50” for “dose response” per reviewer’s suggestion. (Page 2, line 41) accordingly. Page 2, line 41: Keywords: apicomplexa, treatment, cytocrome, new drugs, dose-response. Comments 3: Introduction: The objective should be amended. In the work, the authors propose to investigate the inhibitory assays (PPE) and the percentage of survival. You should include what type of effect you are comparing in the target (“The aim of this study was to compare the effects XXX of….”). Response 3: Thank you for pointing this out. We agree with this comment. The introduction was amended in lines 90 and 91 as: The aim of this study was to compare the effects on B. bovis in vitro survival of ELQ-316, BPQ, ID and the combinations of ID + ELQ-316 and BPQ + ELQ-316. (page 3, lines 90 and 91) Comments 3: Materials and Methods: Statistical analysis Why do you propose a parametric test and not use a non-parametric test knowing that the nature of the data counts. Justify why you did not perform a non-parametric analysis or a generalized linear model (by binary response). If your data counts, you should indicate the media and ranges in your interpretations. Response 3: Thank you for pointing this out. We agree with this comment. We run the non- parametric alternatives of the tests and changes have been made in 2.5 statistical analysis section per reviewer’s suggestion. Results section page 6 to 9. We added a table in the results with media and ranges to better clary this point, (also according to comments below). Comments 4: Results: For lines 166 to 172, it is recommended to include the table to substantiate the results or incorporate it for improved visualization. Response 4: Thank you for pointing this out. We agree with this comment. We added a table to substantiate the results of lines 166 to 172. (page 6, lines 174 to 175) Comments 5: In line 170, the determination of "...but lower speed of kill..." is unclear, as no indicator of half-life is presented. These results only demonstrate dose-response outcomes. It is advisable to revise this expression. Response 5: Thank you for pointing this out. We agree with this comment. We changed the expression per reviewer’s suggestion"...but lower speed of kill..." to the expression: “In contrast, ID and ELQ-316 had similar parasiticidal kinetics, but parasites seemed to have a slower response, at the same drug concentrations.” (page 5, lines 170 to 173) Comments 6: Figure 1: It would be more appropriate to express the concentrations logarithmically, as this would allow for better visualization of the IC50. Does the 1200 point for ID not exhibit variability? It is unclear whether the lines in the figure represent the model or are a smoothed representation connecting the points. It is suggested to include the points and represent the model with shaded confidence intervals. This would enhance the study's visualization. Additionally, it is recommended to organize the labels into individual drugs followed by their combinations. Response 6: Thank you for pointing this out. We agree with this comment. The concentrations were expressed logarithmically. The SD for ID for concentration 1200 nM was 0.69. The figure expresses the survival percentage of each drug concentration and combinations, with the standard deviation (SD). We tried to include the points but it’s too much data that makes the figure confused. The labels have been reordered. (page 7, lines 184) Comments 7: Line 175. The "...sigmoid shape..." is not consistent with the scale presented in your figure. Please revise the figure accordingly. Response 7: Thank you for pointing this out. We agree with this comment. We changed the expression in the text. (page 6, lines 179) Comments 8: Line 188. You state "...high rate of parasite killing...". What comparison are you making? Rate or mean (%)? This point is crucial as it should be supported by your statistical analysis. Response 8: Thank you for pointing this out. We agree with this comment. We eliminated the word “rate” to avoid confusion. (page 8, lines 193) Comments 9: How do you justify presenting information in both figure and table format? This presents similar information. If the authors propose to conduct a comparative dose-response study, which I fully endorse, use the model parameter information to make your comparisons (e.g. IC50). What is the value of partial comparison for each concentration? I suggest removing Tables 1 and 2 and incorporating their discussion into the text. Response 9: Thank you for pointing this out. We agree with this comment. We added tables 1 and 2 as supplementary material. We greatly appreciate the suggestion to use the IC50, in fact we are performing the comparative dose response study to evaluate the level of synergy based on this previous data. Comments 10: Lines 194-209. This type of discussion is unnecessary if the authors interpret the dose-response model. Response 10: We greatly agree, but it was necessary to clarify this point according to other revisions. Comments 11: Table 3: Organize the data from lowest to highest IC50 and include letters to indicate statistical differences. Ensure clarification of the abbreviations. Response 11: Thank you for pointing this out. We agree with this comment. We made the changes according to the suggestions (page 9, lines 230 and 231). Comments 12: Section 3.2 It is advisable to consider including the IC99 in your results. This would enable the determination of concentration at which a cut-off recommendation could be established for comparisons. This is particularly relevant if the intention is to translate the results to vivo studies subsequently. This information could be discussed in Table 4. Response 12: We agree and added the IC99 to table 2 (page 9, lines 232-245). |
|

Reviewer 2 Report
Comments and Suggestions for Authors
This study aimed to evaluate the effects of BPQ, ELQ-316, and their combination on the survival of Babesia bovis in vitro, compared to the conventional drug imidocarb (ID). The results demonstrate that all treatments significantly reduced the survival rate of B. bovis, with the BPQ + ELQ-316 combination showing the lowest IC50 value of 31.21 nM. Notably, the combination also exhibited enhanced efficacy over ELQ-316 alone at lower concentrations.
However, the study has certain limitations. First, the experiments were conducted solely in vitro, indicating the need for further validation in animal models to establish the in vivo relevance of the findings. Second, while the IC50 value of the combination was lower than that of the individual drugs, the fold change was relatively modest, suggesting an additive rather than a synergistic effect. I recommend that the authors provide additional evidence to clarify the potential mechanisms underlying the interaction between BPQ and ELQ-316.
Furthermore, considering that Pharmaceutics is a JCR Q1 journal that typically publishes high-impact, innovative research, this study’s level of innovation may be somewhat limited. While the study holds value, it may be better suited for a journal with a different scope. Therefore, I suggest the authors consider submitting their manuscript to an alternative journal that aligns more closely with their study’s contributions.
Author Response
We appreciate your time in evaluating our work. In the conclusions of the manuscript, we noted that our findings should be tested in an animal model. While we suggested the possibility of synergistic effects, this topic was not analyzed in depth within this paper. In fact, we have investigated the synergistic effect in a subsequent study, and the results will be published in a future manuscript.
Regarding the scope of the journal, we respect Reviewer #2's opinion on this matter. However, we strongly believe that the data presented in this study provides valuable insights for designing future research that could lead to new and improved treatments for bovine babesiosis, a disease with significant economic impact worldwide.
Reviewer 3 Report
Comments and Suggestions for Authors
Infectious diseases transmitted by vectors ticks remain a serious problem for farmers. The manuscript by Cardillo and co-workers explored the effects of combining buparvaquone and ELQ316 in comparison with ELQ316 with imidocarb on Babesia bovis. The study revealed that all drugs as monotherapy or combination therapy showed beneficial effect in B. bovis survival. Although BPQ remain effective drug, the study demonstrated a potential synergy when combined with ELQ-316 during the treatment. Therefore, the study is a significant scholarly contribution in veterinary drug discovery. I will support the acceptance of the paper for publication.
Author Response
We greatly appreciate your revision on our manuscript, comments and support.
Round 2
Reviewer 2 Report
Comments and Suggestions for Authors
The author has diligently addressed the previous round of review comments and has made meticulous revisions to the manuscript. The structure of the paper is now more sound, and the arguments are presented with greater clarity. Although “the absence of animal model research does limit the depth of the study to some extent, it is important to note that the experimental data and conclusions presented in the paper still hold significant innovation and practical value”. Therefore, based on the overall quality of the manuscript, the significance of the research outcomes, and their potential impact on the scientific community, I believe that “the paper meets the publication standards of the journal and I recommend considering its acceptance” .